# Combining the HCT-CI, G8, and AML-Score for Fitness Evaluation of Elderly Patients with Acute Myeloid Leukemia: A Single Center Analysis

**DOI:** 10.3390/cancers15041002

**Published:** 2023-02-04

**Authors:** Semra Aydin, Roberto Passera, Marco Cerrano, Valentina Giai, Stefano D’Ardia, Giorgia Iovino, Chiara Maria Dellacasa, Ernesta Audisio, Alessandro Busca

**Affiliations:** 1Department of Oncology, Hematology, Immuno-Oncology and Rheumatology, University Hospital of Bonn, 53127 Bonn, Germany; 2Department of Oncology, Hematology, A.O.U. Città della Salute e della Scienza, 10126 Turin, Italy; 3Department of Medical Sciences, A.O.U. Città della Salute e della Scienza, University of Torino, 10126 Turin, Italy; 4Department of Hematology, Ospedale Civile, 10073 Ciriè, Italy; 5Department of Oncology, SSD Stem Cell Transplant Center, A.O.U. Città della Salute e della Scienza, 10126 Turin, Italy

**Keywords:** elderly acute myeloid leukemia, frailty scores, G8-score, AML-score for CR, HCT-CI

## Abstract

**Simple Summary:**

In elderly patients with acute myeloid leukemia (AML), non-relapse mortality (NRM) after intensive chemotherapy and, if necessary, allogeneic hematopoietic stem cell transplantation (HSCT) depends on an accurate initial patient fitness assessment. We determined three scores in 197 patients at diagnosis and subsequently investigated the patients’ respective overall survival with each fitness score assignment. The G8-score, the HCT-CI and the AML-score for complete remission (CR) were able to significantly separate “fit” from “unfit” patients, *p* < 0.001, *p* = 0.008 and *p* = 0.001, respectively. Their predictive power was statistically confirmed by univariate analysis and was especially enhanced when combining them. Thus, in the present study, all three scores, simple, fast and, if necessary, also without cytogenetic information, represent significant tools for fitness evaluation before intensive treatment of elderly AML.

**Abstract:**

Background: Accurate assessment of elderly acute myeloid leukemia (AML) patients is essential before intensive induction chemotherapy and subsequent allogeneic hematopoietic stem cell transplantation. In this context, we investigated the capacity of three scores for frailty prediction. Methods: At diagnosis, 197 patients were clinically evaluated for appropriate treatment intensity. In parallel and independently, the G8-score, the Hematopoietic Stem Cell Index (HCT-CI) and the AML-score for CR were determined for each patient and analyzed with respect to overall survival (OS). Results: The G8-score and the HCT-CI were able to significantly separate “fit” from “unfit” patients, <0.001 and *p* = 0.008. In univariate Cox models, the predictive role for OS was confirmed: for the G8-score (HR: 2.35, 95% CI 1.53–3.60, *p* < 0.001), the HCT-CI (HR: 1.91, 95% CI 1.17–3.11, *p* = 0.009) and the AML-score (HR: 5.59, 95% CI 2.04–15.31, *p* = 0.001), the latter was subsequently used to verify the cohort. In the multivariate Cox model, the results were confirmed for the G8- (HR: 2.03, *p* < 0.001) and AML-score (HR: 3.27, *p* = 0.001). Of interest, when combining the scores, their prediction capacity was significantly enhanced, *p* < 0.001. Conclusions: The G8-, the HCTCI and the AML-score represent valid tools in the frailty assessment of elderly AML patients at diagnosis.

## 1. Introduction

The age of 60, although arbitrary and not evidence-based, is internationally held to be the cut-off point between “older” and “younger” patients with acute myeloid leukemia (AML) [1]. The prognosis of AML in the elderly is generally poor. Intensive induction chemotherapy in older patients with AML has an early death (ED) rate of 15–30% [2,3,4] and induces a complete remission (CR) in only about 50% of them [4,5]. The five-year overall survival (OS) rate is around 15% [3].

There are several reasons for these outcome rates. First of all, age itself is an independent, negative prognostic factor in AML [3]. The higher incidence of adverse cytogenetic abnormalities and overexpression of multidrug resistance proteins [1,3,6] determine more frequent resistant disease. More complex comorbidity constellations account for higher non-relapsed mortality (NRM) and impaired overall survival.

It is thus essential to accurately assess patients’ fitness at diagnosis to determine whether they are eligible for intensive chemotherapy and, if necessary, allogeneic hematopoietic stem cell transplantation (HSCT). The indicators that usually rely on performance status, chronological age, and the presence of comorbidities may not be adequate measures for assessing of the heterogeneity of elderly AML. Several studies have devised and proposed scores aiming to predict NRM and patient outcome.

In this context, Palmieri and colleagues retrospectively validated the Ferrara criteria to classify patients’ fitness for intensive chemotherapy [7]. The predictive accuracy resulted in significant short-term mortality, but the prediction of overall survival was less accurate. Further, retrospective and prospective multi-parameter geriatric assessment approaches have been proposed [8,9,10,11]. Apart from producing highly inconsistent results on the role of performance status as a predictor of survival, these approaches resulted in being very time consuming. Currently, geriatric assessments have been incorporated into pre-existing survival prediction models, such as the AML-score [12]. or the Ferrara criteria [7,13]. Such integrated approaches have not become standard because of their complex, time-consuming, multi-parameter assessment modules.

Moreover, fitness scores from different settings, such as the hematopoietic cell transplantation-specific comorbidity index (HCT-CI) [14,15,16,17,18], have shown preliminary usefulness in baseline AML comorbidity-related outcome prediction [19]. However, no standard assessment scales have been validated yet for the baseline assessment of elderly AML patients’ fitness before intensive chemotherapy.

The aim of the present study was to investigate the fitness assessment capacity of two scores, namely the HCT-CI and the G8 score [20,21]. These were originally validated in a non-AML-at-diagnosis setting in elderly AML patients before intensive induction chemotherapy. Their prognostic power for OS was determined independently and verified subsequently by the AML-score for CR [12] before all three scores were combined in a stepwise combination model.

## 2. Materials and Methods

### 2.1. Patients and Treatments

As shown by the CONSORT diagram in Figure 1, a total of 197 ≥60-year-old AML patients were consecutively treated between June 2013 and June 2020 in the hematology department at the Department of Oncology, Hematology, A.O.U. Città della Salute e della Scienza, Turin, Italy.

Acute promyelocytic leukemia patients were excluded. Each patient was evaluated at diagnosis by a hematologist according to general performance status, a clinical examination, and, when available, echocardiography and spirometry. Each case was then referred to a multidisciplinary board for discussion of the most appropriate treatment intensity and type. The same board would then be involved in any later re-evaluation of cases after intensive induction chemotherapy before proceeding to allogeneic HSCT or other changes in treatment. Meanwhile, each patient’s fitness was assessed using three scores: (i) a regional geriatric G8 score [20,21], (ii) the Hematopoietic Stem Cell Index (HCT-CI) [14] and (iii) the AML-score for CR proposed by the German study groups Acute Myeloid Leukemia Cooperative Group and Study Alliance Leukemia [12]. All three scores were obtained at diagnosis and were not known to the multidisciplinary board making treatment decisions.

### 2.2. Frailty Assessment Scores

The G8 score, initially developed to identify newly diagnosed cancer patients for more comprehensive geriatric assessment [20,21], consists of eight items: the patient’s age together with seven other items from the original 18-item Minimal Nutritional Assessment scale [22] (appetite changes, body mass index, weight loss, mobility, neuropsychological problems, medication and self-rated health). The G8 score was calculated from answers to a short questionnaire completed personally by each patient during the first hematology visit. Total scores ranged from 0 to 17, with lower scores indicating a higher risk of impairment. Patients with a G8 score of ≤14 were classified as unfit, while patients with a score of >14 were defined as fit.

Further, the HCT-CI, a tool for capturing comorbidities before allogeneic HSCT, was calculated for each patient [14]. The HCT-CI includes objective laboratory and functional testing data for comorbidities, namely obesity, diabetes, cardiac impairment, including arrhythmia and valve disease, cerebrovascular disease, psychiatric disturbance, renal, hepatic and pulmonary function, rheumatologic disease, peptic ulcer, infections requiring antimicrobial treatment as well as the history of a solid tumor. Patients were stratified as fit with a low- and intermediate-risk HCT-CI score of 0–2 and as unfit with a high-risk HCT-CI score of ≥3, as originally proposed by the authors.

Lastly, the AML-score for CR was calculated for each patient. Standard clinical and laboratory variables (age, body temperature, hemoglobin, platelets, lactate dehydrogenase, fibrinogen and de novo vs. secondary leukemia) as well as the variables, if present at diagnosis, for cytogenetic and molecular risk (low cytogenetic or molecular risk, intermediate cytogenetic risk with an aberrant karyotype, high cytogenetic risk) were inserted into an online form at www.AML-score.org (accessed on 1 April 2022) [12]. The data was used to calculate the percentage values for the probability, after intensive induction chemotherapy, of early death within 60 days and of achieving complete remission. For the sake of consistency, the present study also considered early death to be death from any cause within 60 days of the first day of treatment.

In order to remove the possibility of bias, the scores remained blinded to the multidisciplinary board responsible for making the final decision about the most appropriate therapy regimen for each patient. The study was conducted in accordance with the Declaration of Helsinki and the guidelines for Good Clinical Practice and approved by the Institutional Ethic Committee (00244/2020 in 20 May 2020).

Cytogenetic categorization into favorable-, intermediate- and adverse-risk groups as well as treatment response followed the criteria published on behalf of the European Leukemia Network [23]. Patients with a complete response with incomplete hematologic recovery (CRi) and morphologic leukemia-free state (MLFS) were considered CR patients. The severity of adverse events was graded according to the National Cancer Institute Common Toxicity Criteria (CTCAE, version 4.0).

### 2.3. Statistical Analysis

The primary endpoint was OS, defined as the time from the start of intensive chemotherapy to death from any cause. OS was investigated either by the Kaplan-Meier method (comparing survival curves across groups by the log-rank test) or by the Cox proportional hazards model (comparing the two arms by the Wald test and calculating the 95% confidence intervals). The predictive role on OS was tested for the above-cited scores: HCT-CI- (≥3 vs. 0–2), G8 score (≤14 vs. >14) and the AML-score for CR (high vs. low risk, either by quartiles or as a binary one). In the latter case, its optimal cut-point was estimated by the maximally selected rank statistics, an outcome-oriented method providing an AML-score for the CR cut-off value that corresponds to the most significant relation with OS [24].

Patients’ characteristics were tested with Fisher’s exact test for categorical variables and the Mann–Whitney test for quantitative ones, describing them as median (inter quartile range, IQR). All *p*-values were obtained using the two-sided exact method at the conventional 5% significance level. All data as of May 2022 were analyzed by the R 4.2.0 MaxStat package (R Foundation for Statistical Computing, Vienna-A, http://www.R-project.org (accessed on 1 April 2022).

## 3. Results

### 3.1. Patients and Treatment Characteristics

A total of 197 patients were evaluated by the hematologist and then successively by the multidisciplinary board. Seventy-seven of them were declared to be clinically unfit for intensive chemotherapy and prescribed demethylating agents, low-dose cytarabine or best supportive care (BSC). The remaining 120 patients were evaluated as clinically fit for intensive chemotherapy and treated according to the “7+3” [25] or “FLAI” [26] protocols. In eight FLT3-ITD mutated patients, an FLT-3 inhibitor was added to the intensive induction chemotherapy regimen. Seven of them (87.5%) maintained an FLT-3 inhibitor after allogeneic HSCT. Baseline and treatment characteristics of the intensively treated cohort are summarized in Table 1.

A subgroup of 27 patients (22.5%) required dose adjustments for age and comorbidities. To compare similarly treated patients, only the 120 patients undergoing intensive chemotherapy were included in the analyses. In cases where CR was achieved, consolidation therapy proceeded with intermediate-dose cytarabine and early allogeneic HSCT. In cases of primary induction failure (PIF), remission was re-challenged with the MEC-(mitoxantron 6 mg/m^2^ d1-6, etoposid 80 mg/m^2^ and cytarabine for 6 h 1 g/m^2^ d1-6 i.v.) or the FAM-(fludarabin 30 mg/m^2^ d1-5, cytarabine 2 g/m^2^ d1-5 and mitoxantrone 8 mg/m^2^ d1-3 i.v.) protocols. Of note, most of the intensively treated patients were classified as WHO high-risk patients (59.5%) with ELN-adverse cytogenetics (45.8%) [23].

### 3.2. Toxicity and Overall Response

Nearly half of the patients, n = 54 (44.6%) developed grade III/IV toxicities after induction chemotherapy, mainly consisting of infectious events, new-onset atrial fibrillation and bleeding complications (Table 1). A total of 17 (14%) patients required non-invasive ventilation, with nearly half of them receiving vasopressin support. Only one patient (0.8%) was admitted to the intensive care unit (ICU).

After intensive induction chemotherapy, complete remission was achieved in 73/120 (60.8%) patients, while 40 (33.3%) were refractory. Twenty-five (20%) of the PIF patients were able to complete a re-induction cycle, 13 (10.8%) of whom achieved a second CR (Table 1). The resistant patients were directed to demethylating agents or BSC. During induction, seven patients died due to treatment toxicity or disease progression, marking an early death rate of 5.8%. A total of 33 (27.4%) patients could be successively directed to allogeneic HSCT, with three of them (8.8%) dying due to NRM within 100 days of transplant.

With a median follow-up (FU) of 38.7 months (IQR: 27.2–75.1), the median OS of the clinically fit defined patients undergoing intensive chemotherapy was 15.0 months (IQR: 6.7–50.5). Of note, the median OS of the 77 clinically unfit declared patients who received demethylating agents or BSC was 3.5 months (IQR: 1.2–9.6).

As expected, when stratifying the 120 intensively treated patients according to age, the median OS decreased with age. Patients between 60–65 years of age (n = 43, 35.8%) had a median OS of 20.1 months (IQR: 6.7—not reached) compared to 66–70-year-old patients (n = 45, 37.5%) with 14.6 months (IQR: 7.4—not reached) and ≥71-year-old patients (n = 32, 26.7%) with a median OS of 12.6 months (IQR: 6.1–22.7), *p* = 0.147 (Appendix A).

### 3.3. Fitness Score Evaluation in the Intensively Treated Patient Subgroup

Blinded to the baseline clinical evaluation, each patient’s G8 score along with their HCT-CI and AML-score for CR were obtained at diagnosis and successively associated with their OS. The G8 score significantly distinguished 65 (54.2%) fit from 55 (45.8%) unfit patients. G8-fit classified patients had a significantly longer median OS of 20.7 months (IQR: 11.5—not reached) than the G8-unfit classified ones with a median OS of 7.8 months (IQR: 4.4–20.6), *p* < 0.001 (Figure 2A). So did the HCT-CI: HCT-CI-fit defined patients (n = 94, 78.3%) showed a significantly longer median OS of 16.1 months (IQR: 7.8—not reached) than the HCT-CI unfit defined ones (n = 26, 21.7%) with a median OS of 6.7 months (IQR: 2.9–18.6), *p* = 0.008 (Figure 2B).

Given that the initial selection of fit patients depended on the clinical decision made by the hematologist/multidisciplinary board, a selection bias was evident. Therefore, the cohort was further analyzed using the AML-score, previously published by the German Acute Myeloid Leukemia Cooperative Group (AMLCG) and Study Alliance Leukemia (SAL), based on two large cohorts of elderly AML patients in the same setting. Krug et al. hypothesized that CR achievement is a prerequisite for long-term survival [12]. Following the same rationale and approach, the predicted CR probability was used to stratify fit from unfit patients in our study as well, and the predicted CR rates were divided arbitrarily into quartiles. In the present cohort, the predicted CR probability ranged from 19.0% to 79.8%. The first quartile was defined as ≤37.2%, the second as 37.3–45.2%, the third between 45.3–54.6% and the fourth quartile ≥54.7%. In fact, the AML-score for CR separated significantly in terms of OS, *p* = 0.006 (Figure 2C). The higher the predicted CR according to THE AML-score, the higher was the median OS, ranging from 9.0 months (IQR: 2.7–21.9, n = 39) in the first quartile to 10.8 months (IQR: 5.0–18.6, n = 37) in the second, 15.0 months (IQR: 7.6–38.1, n = 27) in the third, and 39.6 months (IQR: 8.8—not reached, n = 17) in the fourth quartile.

The AML-score for CR represents a continuous variable, while the clinical decision, G8 score and HCT-CI are binary variables (fit/unfit). For this reason, a cut-point selection analysis using the maximally selected rank statistics was performed [24]. The ideal binary cut-off for the predicted CR probability proved to be 63.2%. Indeed, when stratifying the patients according to this dichotomous cut-off, the difference was highly significant, *p* < 0.001 (Figure 2D); the median OS for the fit patients (4/16 died, 25%) was not reached, while the median OS for the unfit patients (82/104 died, 78.8%) was 12.1 months (IQR: 6.1–29.0).

Next, to analyze the ability of each of the three scores to independently predict OS, a series of univariate Cox models was performed. The predictive role of all three scores for OS was highly significant. The G8 score resulted in a HR of 2.33 (95% CI 1.68–3.25, *p* < 0.001), whereas the HCT-CI resulted in a HR of 1.78 (95% CI 1.29–2.46, *p* < 0.001) and the AML-score for CR resulted in a HR of 4.07 (95% CI 1.99–8.32, *p* < 0.001).

While all three scores had a significant impact on OS, when analyzed by a series of univariate Cox models, the multivariate one confirmed the OS prognostic role only for G8 (HR: 2.03, *p* < 0.001) and AML (HR: 3.27, *p* = 0.001). The four survival models are reported in Table 2.

After having shown the prognostic capacity of each single score, the next step was to analyze their combined capacity in a hypothetical model. Four subgroups were created by creating a first subgroup containing only patients with no unfavorable scores and adding another unfavorable score to each successive subgroup, such that the fourth subgroup contained patients with three unfavorable scores. The OS of each of the groups was then compared. The combined capacity of the three scores proved to be highly statistically significant. As the number of unfit scores increased, the OS in the respective subgroup decreased. Patients with all three unfavorable scores (15.8%) had a median OS of 6.2 months (IQR: 2.7–20.7), while patients with two unfavorable scores (34.2%) had a median OS of 8.2 months (IQR: 5.4–17.5). Patients with just one unfavorable score (38.3%) had a median OS of 16.2 months (IQR: 11.5—not reached), while for patients with no unfavorable scores (11.7%), the median OS was not reached (Figure 3, *p* < 0.001).

### 3.4. Fitness Score Evaluation in the Non-Intensively Treated Subgroup

The remaining 77 clinically unfit evaluated patients received demethylating agents, low-dose cytarabine or best supportive care. In this subgroup, the G8 score defined 61 (79.2%) patients as unfit and 16 (20.8%) as fit, while the HCT-CI assessed 41 (53.2%) as fit and 36 (46.8%) patients as unfit. Of the clinically unfit patients defined by the AML score for CR, using the same cut-off value of 63.2% as a binary variable identified 74 (96.1%) patients as unfit and 3 (3.9%) patients as fit.

## 4. Discussion

In the present study, the G8 score and HCT-CI were calculated for each patient at diagnosis before intensive chemotherapy and resulted in an assignment of fit or unfit. Stratification by the scores was conducted in parallel and independently from the clinical treatment decision. Consequently, patients’ OS was associated with fitness assignment. Each single score, when considered independently, significantly distinguished fit from unfit patients: the median OS of the G8- as well as of the HCT-CI-fit patients was significantly longer than that of those categorized as unfit. Given that the scores were applied to a clinically selected patient cohort, the AML-score for CR, also determined at diagnosis, was applied to “validate” the cohort. Indeed, in the clinically selected population, fitness prediction by the AML-score in terms of overall survival also proved to be highly significant. The median OS for the AML-score-fit patients was not reached, while the median OS for the AML-score-unfit patients was 12.1 months.

The G8 score as well as the HCT-CI are validated tools developed for patient selection in different clinical settings. The G8 questionnaire, with a mean completion time of 5 min is a fast, simple, and feasible screening tool for identifying older cancer patients at risk of life-threatening events. The ONCODAGE Prospective Multicenter Cohort Study showed good sensitivity and independent prognostic value of the G8 score on 1-year survival of oncological patients [21]. In the present cohort, the discriminating capacity of the G8 score in terms of OS was confirmed in univariate analysis and subsequently even in multivariate analysis. To our knowledge, this is the first study to evaluate the G8 score in a setting of elderly AML patients with a median follow-up period of more than three years.

In addition to the G8 score, the HCT-CI also significantly distinguished fit from unfit patients in terms of OS prediction. The HCT-CI was initially developed by Sorror and colleagues as a tool for capturing pre-transplant comorbidities and was used in predicting the outcome of allogeneic HSCT [14]. Subsequently, it has been consolidated with various disease-specific and patient-specific risk factors to refine assignment to the appropriate HCT setting. Transferring pre-transplant comorbidity scores to an earlier stage of the diagnosis was previously addressed. Etienne and colleagues [27] had retrospectively used an adapted version of the original Charlson Comorbidity Score (CCI) [28] for NRM assessment in elderly AML. They showed that the risk of mortality and probability of OS were significantly associated with the CCI score. Subsequently, Giles and colleagues compared the CCI with the HCT-CI in elderly AML patients at diagnosis and reported that the HCT-CI may be predictive for early death and overall survival [19]. The median ages of the populations described by Etienne and Giles were older than in the present cohort, at 73 and 70 years vs. 68 years, respectively, whereas the median age of the original pre-transplant HCT-CI cohort [14] was significantly younger at 43 years. Sorror himself retrospectively validated the prognostic impact of his score on 1-year mortality using an augmented HCT-CI and including age and cytogenetic/molecular risks in AML at diagnosis in patients 20–89 years old [29]. Despite the differences of the analyzed populations and the data available, in the present cohort, baseline HCT-CI was significantly associated with OS prediction in elderly AML. This supports the hypothesis that the HCT-CI may also have a predictive role outside the allogeneic HSCT setting. In the reported studies of Giles and Etienne, as in the Sorror-cohorts, score stratification was performed retrospectively. In the present study, all scores were applied at diagnosis before the initiation of treatment, thus excluding bias in the clinical assessment.

It was found that not just the individual scores independently significantly distinguished fit from unfit patients, but that the combination of the G8 score, the HCT-CI and the AML-score for CR also resulted in highly significant results, as analyzed in a hypothetical model for OS prediction. The combination of the AML-score with multi-parametric geriatric assessments was recently investigated [13]. Min and colleagues showed, as in our study population, that in a clinically selected cohort for intensive chemotherapy, functional tests, such as the gait speed test and the sit and stand speed test may be significant tests for OS prediction. Combining them with the AML-score for CR enhanced their predictive power, although not all of their combinations did so: the performance status and geriatric depression scales failed to reach statistical significance.

However, in the cohort of Min et al., a relatively high 58% of patients underwent allogeneic HSCT, as opposed to 27.4% in our study. This may be due to the stringent baseline selection for intensive chemotherapy in the Min cohort; patients with active infection or impaired organ function were excluded from intensive chemotherapy. In patients with highly active disease, organ dysfunction might result in seriously compromised performance status at diagnosis, not reflecting the patients’ usual fitness [30]. Therefore, patients with stable organ dysfunction or active infections, including pulmonary insufficiency with non-invasive ventilation support, were not excluded from intensive chemotherapy at our institution.

Cytogenetics are currently considered the most powerful prognostic factor, although recent trials have reported their limitations in elderly AML [29,30]. The genomic landscape of elderly AML patients differs significantly from that of younger patients [31,32,33]. Initial models combining cytogenetic risk with the mutational status of distinct genes, such as *KRAS*, *ASXL1*, *RUNX*, *SETBP1*, *DNMT3A* and *CSF3R*, are proving to be more discriminative than the ELN 2017 classification in risk stratification [32,34,35]. Therefore, the need for a detailed genetic characterization with early identification of targetable mutations is becoming of pivotal importance. A plethora of emerging targeted drugs with good tolerability profiles suggests that it might be possible for patients to achieve CR and proceed subsequently to allogeneic HSCT consolidation. For instance, favorable outcomes are reported from combining venetoclax with hypomethylating agents, especially in *NPM1*-mutated cases [36] or ivosidenib with azacytidine in *IDH1*-mutated AML [37]. Even triple combinations with low-dose cytarabine, azacytidine and cladribine [38] or venetoclax, decitabine and FLT3-inhibitors [39,40] are currently under investigation as front-line therapy for unfit elderly with AML. Obtaining significant CR rates in initially unfit patients may allow them to be re-evaluated later for allogeneic HSCT. These approaches may completely modify the concept of fit/unfit in elderly AML patients. Together with Urbino and colleagues, we went even further and discussed whether patients with unfavorable genetics should only be considered for intensive induction chemotherapy if they are potentially eligible for successive allo-HCT, as results with chemotherapy alone are usually dismal [41,42].

Although several studies have shown that waiting for cytogenetics [43,44] does not have an impact on induction treatment outcome, many hematologists feel compelled to start treatment before cytogenetics data is available. Scarce material quality or an incomplete metaphase quantity may also represent reasons for the unavailability of cytogenetic information at the initiation of treatment. The G8 score as well as HCT-CI can be determined without cytogenetic or molecular data. The AML-score for CR, although more accurate with cytogenetics, can also be obtained without cytogenetics or molecular characteristics. The present results underline that all three scores may serve independently as tools for “fitness” assessment in elderly AML patients. Their combination is even more significant and increases their predictive power.

Our study has several limitations. The selection of OS instead of early death as the primary endpoint may be objected to when addressing patients’ fitness for intensive chemotherapy. In our center, the early mortality rate at 60 days (5.8%) was considerably low. As already underlined by Krug and colleagues, because CR after intensive induction chemotherapy is a prerequisite for long-term survival in AML, the AML-score for CR was used as a surrogate marker for OS prediction [12].

As a single-center study, only patients who were considered fit for intensive induction chemotherapy were included. This selection bias was addressed by analyzing the selected population according to the AML-score established in a larger cohort in the same setting. Another limitation may rest in the two different induction chemotherapy protocols (“7+3”-or FLAI protocol) used, although several studies have shown that the chosen intensive treatment regimens at the time of patient acquisition do not greatly affect outcome in elderly AML patients [3,45]. The present study does not formally represent a prospective study, although all three scores were obtained at diagnosis and were not influenced by the clinical treatment decision.

## 5. Conclusions

In conclusion, the G8 score, the HCT-CI and the AML-score for CR are quick to obtain even without cytogenetics and represent significant tools in the baseline fitness assessment of elderly AML before intensive chemotherapy. Their predictive power for OS is enhanced even when using them in combination. Given that confirmation data is lacking, an individual assessment of each case remains mandatory. Prospective trials will investigate their role in the context of novel intensive chemotherapy regimens incorporating targeting molecules, especially within the recently published ELN 2022 classification [46].

## Figures and Tables

**Figure 1 cancers-15-01002-f001:**
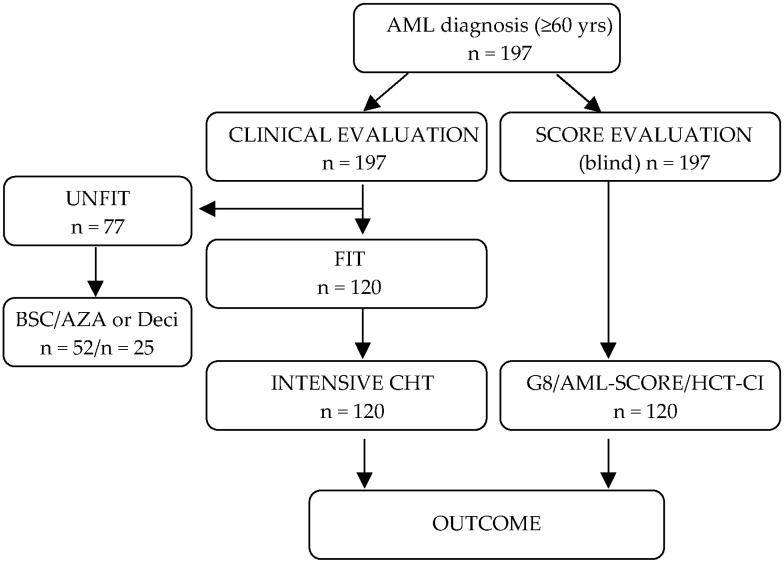
Consort diagram of the study population. All patients were evaluated at diagnosis clinically by a hematologist and subsequently by a multidisciplinary board, where treatment decision was made. In parallel and blinded, each patient’s fitness was assessed by the three scores. 120/197 were evaluated as fit and underwent intensive chemotherapy. Although the scores were applied to all patients, only the patients undergoing intensive chemotherapy were considered for analysis. BSC indicates best supportive care; AZA, azacytidine; Deci, decitabine; CHT, chemotherapy.

**Figure 2 cancers-15-01002-f002:**
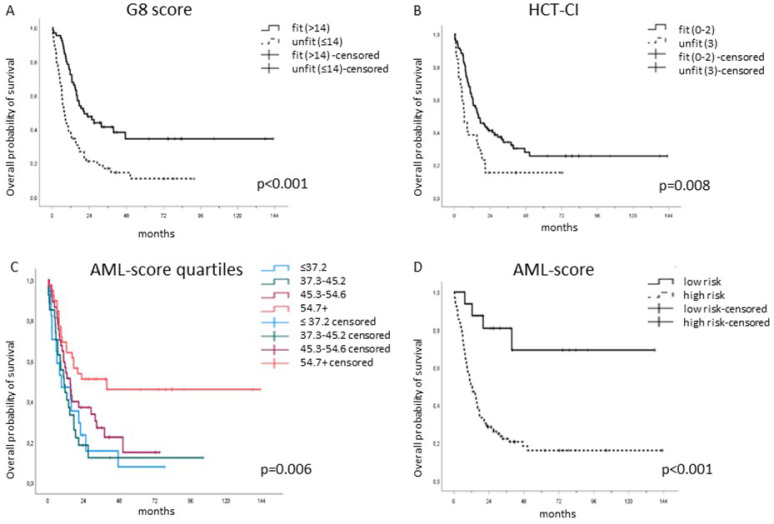
The G8, HCT-CI- and AML-score for CR separated significantly fit from unfit patients. (**A**) G8-fit classified patients (n = 65, 54.2%) had a median OS of 20.7 months (IQR: 11.5—not reached) compared to the G8-unfit classified ones (n = 55, 45.8%) with a median OS of 7.8 months (IQR: 4.4–20.6), *p* < 0.001. (**B**) HCT-CI-fit defined patients (n = 94, 78.3%) showed a median OS of 16.1 months (IQR: 7.8—not reached) compared to the HCT-CI-unfit ones (n = 26, 21.7%) with a median OS of 6.7 months (IQR: 2.9–18.6), *p* = 0.008. (**C**) The AML-score for CR was able to separate significantly fit from unfit patients as an ordinal variable, *p* = 0.006 as well as a (**D**) binary variable, *p* < 0.001.

**Figure 3 cancers-15-01002-f003:**
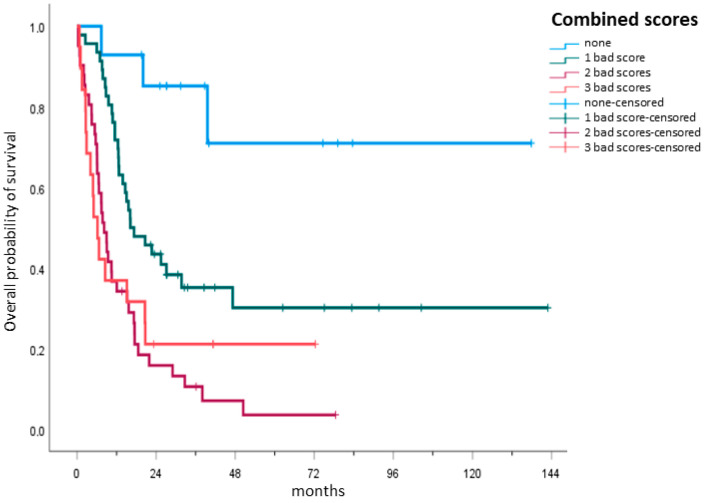
Hypothetical combination model of the scores. A hypothetical model was used to analyze the prediction capacity of all three scores. Four subgroups were created, with the first subgroup containing only patients with no unfavorable or “bad” scores and adding another unfavorable score to each successive subgroup. Patients with all three unfavorable scores (n = 19, 15.8%), defining them as unfit, had a median OS of 6.2 months (IQR: 2.7–20.7), while patients with two unfavorable scores (n = 41, 34.2%) had a median OS of 8.2 months (IQR: 5.4–17.5). Patients with just one unfavorable score (n = 46, 38.3%) had a median OS of 16.2 months (IQR: 11.5—not reached), while for patients with no unfavorable scores (n = 14, 11.7%), the median OS was not reached.

**Table 1 cancers-15-01002-t001:** Baseline, treatment and response characteristics of intensively treated patients.

Variable	n (%)
Patients	120
Median age, years (IQR)	68 (60–77)
Sex (male)	72 (60)
WBC > 100,000 × 10^6^/µL	12 (10)
**WHO n = 120**	
de novo AML	48 (40)
t-AML	9 (7.5)
MDS related	57 (47.5)
Myeloid sarcoma	6 (5)
**Molecular markers, n = 113**	
AML/ETO, CBF, inv16 or t(8; 21)	2 (1.7)
NPM mut	27 (22.5)
FLT3-ITD/FLT3-TKD mut	24 (20)
MLL mut	13 (10.8)
**Cytogenetics, n = 97**	
Normal karyotype	45 (37.2)
Complex karyotype	20 (16.7)
Monosomy	16 (13.3)
del(5q), abn(17p)	19 (15.8)
**Fitness scoring**	
G8, fit (>14)	65 (54.2)
Sorror (0–2)	94 (78.3)
**Treatment regimen**	
7+3	93 (77.5)
FLAI	27 (22.5)
Allogeneic HSCT	33 (27.4)
**Grade 3/4 toxicity**	54 (45)
Infection	27 (22.5)
Atrial fibrillation	4 (3.3)
Bleeding	5 (4.1)
Cerebral ischemia	3 (2.5)
**Treatment response**	
Death in induction	7 (5.8)
CR1	73 (60.8)
PIF	40 (33.3)
CR2	13 (10.8)

Data are indicated as n (%) or median. Percentages refer to a total number of events scored. Because of rounding, percentages do not always add up to 100. n indicates number; WBC, white blood cells; WHO, World Health Organization; AML, acute myeloid leukemia; t-AML, therapy-related acute myeloid leukemia; ETO, ETO gene; CBF, core binding factor; NPM1, nucleophosmin 1; mut, mutated; FLT3-ITD, FMS-like receptor tyrosine kinase-3 internal tandem duplication, TKD, FLT3-tyrosine kinase domain; MLL, mixed lineage leukemia; del, deletion; abn, abnormal; G8, G8 score; 7+3, 7+3-protocol; FLAI, Flag-Ida-protocol; HSCT, hematopoietic stem cell transplant; CR1, first complete remission; PIF, primary induction failure; CR2, second complete remission.

**Table 2 cancers-15-01002-t002:** Survival models for the three scores.

	Univariate Models		Multivariate Model	
	HR (95% CI)	*p*	HR (95% CI)	*p*
HCT-CI (unfit vs. fit)	1.78 (1.29–2.46)	<0.001	1.20 (0.85–1.70)	0.305
G8 (unfit vs. fit)	2.33 (1.68–3.25)	<0.001	2.03 (1.46–2.84)	<0.001
AML (high vs. low risk)	4.07 (1.99–8.32)	<0.001	3.27 (1.59–6.73)	0.001

## Data Availability

The data presented in this study are available in this article and Appendix A.

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
