# Peer review of "Combining the HCT-CI, G8, and AML-Score for Fitness Evaluation of Elderly Patients with Acute Myeloid Leukemia: A Single Center Analysis"

_cancers, 2023, doi:10.3390/cancers15041002_

Round 1

Reviewer 1 Report

In elderly acute myeloid leukemia (AML) patients, accurate assessment of patient’s fitness is increasingly important in decision-making of appropriate interventions. In a clinical study, the authors investigated the capacity of three scores determined at diagnosis for fitness evaluation in 197 elderly AML patients (age ≥ 60 years), including HCT-CI, G8-score, and AML-score. They investigated patients’ respective overall survival (OS) with each score fitness assignment and found that they were all significant tools for fitness evaluation before intensive treatment of elderly AML. They also found that when combining the scores their prediction capacity for OS significantly increased. This study is well written and has a great interest for clinical practice. I have the following comments:

1. According to Figure 1, of the 197 patients 120 were evaluated fit and 77 were unfit by clinical evaluation. Then how many patients were evaluated fit and unfit by each of the three scores, respectively? Could the authors provide these data?

 2.   Although the study included 197 patients and they all received the score evaluation (according to Figure 1), the investigation of the scores prediction for patient’s fitness (determined by their prediction for OS) was actually only performed on the 120 patients who were evaluated fit by clinical evaluation and underwent intensive chemotherapy. Although clinical evaluation is not a perfect evaluation of patient’s fitness, it is reasonable to suppose that the evaluation of the scores prediction in this study was mainly done in fit AML patients. Could the authors provide the score evaluation data on the 77 patients who were considered unfit by clinical evaluation? For instance, what was the proportion of those 77 patients who were considered unfit by HCT-CI, G8-score, and AML-score, respectably? I think adding such information would provide a more complete picture of the capacity of the scores for fitness assessment.

 3.     All three scores, HCT-CI, G8-socre and AML-score, were found to be significant tools for fitness evaluation among elderly AML patients. Since all the original scores are continuous variables, I am wondering if there are any correlations between them. It would be interesting to know.

 4.      Similarly, when the three scores each were used as a binary variable to assess the patient’s fitness, it would be helpful to learn their assessing agreement among the three scoring systems. For instance, when compared to AML-score, what are the positive percent agreement and negative agreement for HCT-CI, and for G-8 score, respectively?

 5.      In addition to univariate analysis, multivariate Cox regression analysis was performed to investigate the independent capacity of the scores for OS prediction. However, very little information was provided on the multivariate analysis. For instance, was the multivariate Cox regression analysis performed for each score separately, or all the three scores were included in a single Cox regression model? Besides, what other variables were included in the multivariate Cox regression analysis, and in addition to G8-score and AML-score, were there any other variables found to be significantly associated with OS by multivariate Cox regression analysis?  

1    6.   According to the authors, early death (ED) was not selected as the primary endpoint in the study because of the low early mortality rate. Did the authors assess the scores prediction for other endpoints such as complete remission (CR) and non-relapse-mortality (NRM)?

 7.      Line 19 – Please provide the full name of CR as the authors did for AML and NRM (Line 15)

 8.      Figure 1 – Please add a footnote providing the full name of “BSC/AZA” and “Deci” for the 77 unfit patients.

 9.      Line 225 – The AML-score quartiles in Figure 2C should be an ordinal variable, not a “continuous” variable.  Please correct it.  

Author Response

Reviewer 1

In elderly acute myeloid leukemia (AML) patients, accurate assessment of patient’s fitness is increasingly important in decision-making of appropriate interventions. In a clinical study, the authors investigated the capacity of three scores determined at diagnosis for fitness evaluation in 197 elderly AML patients (age ≥ 60 years), including HCT-CI, G8-score, and AML-score. They investigated patients’ respective overall survival (OS) with each score fitness assignment and found that they were all significant tools for fitness evaluation before intensive treatment of elderly AML. They also found that when combining the scores their prediction capacity for OS significantly increased. This study is well written and has a great interest for clinical practice. I have the following comments:

  1. According to Figure 1, of the 197 patients 120 were evaluated fit and 77 were unfit by clinical evaluation. Then how many patients were evaluated fit and unfit by each of the three scores, respectively? Could the authors provide these data?

The data on the score evaluation of the 120 clinically fit evaluated patients undergoing intensive chemotherapy is provided in the results part 3.3 from line 208ff. The results concerning the 77 patients were added as a separate paragraph at the end of the results part (line 277ff).

  1. Although the study included 197 patients and they all received the score evaluation (according to Figure 1), the investigation of the scores prediction for patient’s fitness (determined by their prediction for OS) was actually only performed on the 120 patients who were evaluated fit by clinical evaluation and underwent intensive chemotherapy. Although clinical evaluation is not a perfect evaluation of patient’s fitness, it is reasonable to suppose that the evaluation of the scores prediction in this study was mainly done in fit AML patients. Could the authors provide the score evaluation data on the 77 patients who were considered unfit by clinical evaluation? For instance, what was the proportion of those 77 patients who were considered unfit by HCT-CI, G8-score, and AML-score, respectably? I think adding such information would provide a more complete picture of the capacity of the scores for fitness assessment.

See our previous comment.

  1. All three scores, HCT-CI, G8-socre and AML-score, were found to be significant tools for fitness evaluation among elderly AML patients. Since all the original scores are continuous variables, I am wondering if there are any correlations between them. It would be interesting to know.

Right-censored time-to-event data like the ours are not structured for computing correlations and/or agreement assessment. Indeed, this topics would require specific, special calculus techniques. It could be possible to non-parametrically estimate a Spearman‘s correlation coefficient between two time-to-event variables (first computing Dabrowska’s bivariate survival surface from time-to-event data and then using it to calculate Spearman’s  correlation coefficient). But we are dealing with three scores, and not two: thus, any answer would be unconclusive, at the light of multiple comparison problem. Censored observations usually do not require such an approach, since these answers can be more precisely derived from a multivariate Cox proportional models.

  1. Similarly, when the three scores each were used as a binary variable to assess the patient’s fitness, it would be helpful to learn their assessing agreement among the three scoring systems. For instance, when compared to AML-score, what are the positive percent agreement and negative agreement for HCT-CI, and for G-8 score, respectively?

See our previous comment.

  1. In addition to univariate analysis, multivariate Cox regression analysis was performed to investigate the independent capacity of the scores for OS prediction. However, very little information was provided on the multivariate analysis. For instance, was the multivariate Cox regression analysis performed for each score separately, or all the three scores were included in a single Cox regression model? Besides, what other variables were included in the multivariate Cox regression analysis, and in addition to G8-score and AML-score, were there any other variables found to be significantly associated with OS by multivariate Cox regression analysis?

We warmly thank the reviewer for this comment. Even if at time of last follow-up 158/197 patients (80.2%, died of the whole cohort, so no problem with a low number of OS events), we decided to skip the classical hematological risk factors and to focus OS modeling only on the survival impact of the three scores, treated as OS determinants. The four survival models are now reported in a dedicated table. Additionally, the information was included in the text from line 252f.

Table 2. Survival models of the three scores

univariate models

multivariate model

HR (95%CI)

p

HR (95%CI)

p

HCT-CI (unfit vs fit)

1.78 (1.29-2.46)

<0.001

1.20 (0.85-1.70)

0.305

G8 (unfit vs fit)

2.33 (1.68-3.25)

<0.001

2.03 (1.46-2.84)

<0.001

AML (high vs low risk)

4.07 (1.99-8.32)

<0.001

3.27 (1.59-6.73)

0.001

  1. According to the authors, early death (ED) was not selected as the primary endpoint in the study because of the low early mortality rate. Did the authors assess the scores prediction for other endpoints such as complete remission (CR) and non-relapse-mortality (NRM)?

We have chosen OS as the main endpoint, because we wanted to focus on the “predictive capacity” of the scores for frailty. The main aim of the study was investigating patients’ fitness for intensive chemotherapy, in first line independently of their response. In fact we did not analyze their association with response or NRM.

  1. Line 19 – Please provide the full name of CR as the authors did for AML and NRM (Line 15)

Corrected in line 19

  1. Figure 1 – Please add a footnote providing the full name of “BSC/AZA” and “Deci” for the 77 unfit patients.

Corrected in the figure legend.

  1. Line 225 – The AML-score quartiles in Figure 2C should be an ordinal variable, not a “continuous” variable.  Please correct it.  

Corrected in the figure legend.

Reviewer 2 Report

The authors tested the clinical significance of  G8 score,  HCT-CI, and, AML-score for risk and overall survival prediction of elderly AML patients, and validated the importance of the combination of these prognostic scores.

Minor things need to be supplemented.

1)  HCT-CI is needed to be described in the manuscript.

2) medical record of the patient is included in the HCT-CI scoring, and these factors are overlapped with G8 score.

3) HCT-CI score was originally classified into 3 groups, >3, 1-2, and 0, why did you categorize it into 2 groups, >3, =<2? 

4) is there any bias in treatment?

for example, unfit or patients with high bad scores were treated with non-intensive CTx and it may affect the outcome.

Author Response

  • HCT-CI is needed to be described in the manuscript.

The description of the HCTCI was added in line 116ff.

  • medical record of the patient is included in the HCT-CI scoring, and these factors are overlapped with G8 score.

Both scores include psychiatric impairment and the BMI. In the G8 score both, BMI and neuropsycholgical problems, are represented in different clinical gradings/severity. The HCT-CI considers only the presence of a BMI of >35 kg/m2 as a risk factor and counts a psychiatric disturbance when a depression or an axiety requires psychiatric consult or treatment.

HCT-CI score was originally classified into 3 groups, >3, 1-2, and 0, why did you categorize it into 2 groups, >3, =<2? 

The survival and NRM risk curves in the original study from Sorror and colleagues show that 0 and 1-2 patients are similar compared to the curves from HCT-CI ≥ 3 patients. In daily practice HCT-CI ≥3 pre-transplant is considered as a high risk factor. In concordance with daily practice and in order to have a binary variable, we categorized 0 with 1-2 versus >=3.

4) is there any bias in treatment?

for example, unfit or patients with high bad scores were treated with non-intensive CTx and it may affect the outcome.

As mentioned in line 129ff the scoring was blinded to the hematologist and the interdisciplinary board leading to treatment decision. Further, in order to exclude a treatment bias as mentioned in line 176 ff., only intensively treated patients were included in the analysis.

In line 141 “OS rate” the word rate was cancelled.

In line 147 the word optional was replaced by “optimal”.

In line 83 the word “consort” was replaced by “CONSORT”.

In line 146 the word “HCTCI” was replaced by “HCT-CI”.

Reviewer 3 Report

I have reviewed the manuscript. This is a nicely performed study and a well written paper. I have a few comments.

1) Sharing the comparison of the scores and OS between patients who transplanted and those who did not receive an allograft would be interesting.

2) Did patients with FLT3-ITD/TKD positivity receive FLT3 inhibitors? If yes, these should be shared and discussed.

3) I think the same scores should be tested in patients planning to receive HMA/LDAC+VEN combinations. I am interested to hear the comment of the authors on this as well.

4) So, what do the authors suggest? Should patients with high scores be given low intensity treatments, etc.? This can be commented in the conclusion part. 

Author Response

Reviewer 3

I have reviewed the manuscript. This is a nicely performed study and a well written paper. I have a few comments.

  • Sharing the comparison of the scores and OS between patients who transplanted and those who did not receive an allograft would be interesting.

A comparison of the scores and OS between transplanted and non-transplanted patients would be of high scientific interest. In the present cohort a total of 33 patients underwent allogeneic HSCT. Unfortunately this sub-cohort is representing a too small cohort size for efficient analysis of all respective scores.

  • Did patients with FLT3-ITD/TKD positivity receive FLT3 inhibitors? If yes, these should be shared and discussed.

This information was added in line 164.

  • I think the same scores should be tested in patients planning to receive HMA/LDAC+VEN combinations. I am interested to hear the comment of the authors on this as well.

As we discussed from line 353ff and in line 394ff investigation of the scores in the context of HMA/LDAC+VEN would be of high clinical interest. Generally, an initially unfit patient may receive HMA/LDAC+VEN and once achieving CR and enhancing fitness may be re-assessed for successive intensive treatment, such as reduced intensity conditioned allogeneic HSCT. In this context a re-assessment by the same scores may have a role.

  • So, what do the authors suggest? Should patients with high scores be given low intensity treatments, etc.? This can be commented in the conclusion part. 

The results indicate that patients with high risk scores may be more fragile for intensive induction chemotherapy regimens. Given that prospective multi-center confirmation data is lacking, omission of intensive treatment to high risk patients is not clearly justified yet. An individual assessment of each case remains mandatory and, when supposed to be beneficial, dose adjustments for high risk patients‘ fitness may be considered.

A comment was added in the conclusion part in line 394ff.

In line 141 “OS rate” the word rate was cancelled.

In line 147 the word optional was replaced by “optimal”.

In line 83 the word “consort” was replaced by “CONSORT”.

In line 146 the word “HCTCI” was replaced by “HCT-CI”.
